# A DFT Study on the Interaction of Doped Carbon Nanotubes with H$_2$S, SO$_2$ and Thiophene

Hossein Tavakol [1,*] and Hamed Haghshenas [2]

1 Department of Chemistry, Isfahan University of Technology, Isfahan 84156-83111, Iran
2 Division of Biochemistry, Department of Biology, Faculty of Sciences, Shahrekord University, Shahrekord 88186-34141, Iran; hamed.haghshenas@stu.sku.ac.ir
* Correspondence: h_tavakol@cc.iut.ac.ir; Tel.: +98-3133913241

**Abstract:** The interactions of simple and Al-, B-, N-, S-, P-, and Si-doped carbon nanotubes with three sulfur-containing molecules (H$_2$S, SO$_2$, and thiophene) were investigated to assess their adsorption potencies and sensor abilities. The DFT method was used to calculate the adsorption energies and natural bond orbitals parameters. In addition, population analyses were performed to calculate the energy gaps and reactivity parameters. The results showed an exothermic interaction of H$_2$S, SO$_2$, and thiophene with simple and doped carbon nanotubes, while the maximum negative adsorption energies belong to Al- and B-containing complexes. Furthermore, evaluation of second-order perturbation energies (obtained from natural bond orbitals calculations) confirmed that the highest energies were related to B- and Al-containing intramolecular interactions. The results revealed the favorability of adsorption of SO$_2$ by nanotubes (B- and Al-doped carbon nanotubes, in particular) compared with the other examined adsorbates.

**Keywords:** adsorption; doped; nanotube; sulfur; sensor





## 1. Introduction

Based on its unique structural properties and wide range of applications, carbon nanotubes (CNTs) have attracted broad interest from various research groups [1–4] since their first report in 1991 [5]. The adsorption abilities of CNTs provide an excellent opportunity to solve environmental pollution problems and to prepare a new category of useful sensors. In recent years, lots of studies—tens of thousands (experimental or theoretical)—have been performed to evaluate the adsorption properties of CNTs and their doped derivatives [6–12]. In particular, the theoretical reports showed an increase in stability and hydrogen adsorption capacity of the CNTs in the presence of dopant atoms [13,14].

Moreover, the energy gaps analysis of carbon nanotubes, doped by various heteroatoms, proved that different heteroatoms could have different effects on the conductivities of nanotubes [15]. Consequently, electrical conductivity and chemical reactivity of nanotubes could be improved by doping with heteroatoms [16], which make them proper candidates for application in chemical sensors. Recently, the application of doped CNTs as a group of pollutant absorbents was extensively studied. In this line, recent experimental works showed that simple and doped carbonaceous materials could be used to adsorb sulfur-based environmental pollutants, including H$_2$S, SO$_2$, and thiophene [17].

Despite all the reported studies related to the adsorption and sensor properties of doped carbon nanotubes, there are only a few reports considering the effect of various heteroatoms on these properties [18]. Therefore, a comprehensive study on the adsorption of desired molecules on the surface of carbon nanotubes is still required. In this regard, the molecular properties, interaction energies, and sensor properties of CNTs in the presence of sulfur-containing pollutants should be investigated. For this purpose, hydrogen sulfide, sulfur dioxide, and thiophene were selected as sample molecules for common sulfur-containing small molecules. The toxicities of sulfur-containing compounds, especially

hydrogen sulfide, sulfur dioxide, and thiophene have been studied extensively [19]. These compounds can be found in crude oil and their combustion products can be released into the air, leading to several environmental issues such as acidic rain. Therefore, studies in the detection and separation of these molecules are environmentally quite important.

Moreover, hydrogen sulfide inhibits the activity of some biological enzymes such as cytochrome oxidase and its high concentration quickly causes death [20]. In addition, several researchers have reported the effect of sulfur dioxide on asthma, bronchitis, and mortality. Thiophene can also be the reason for the degeneration of neurons in the inferior colliculus and the cerebral cortex [21]. Therefore, in the course of our interest in the adsorption and sensor properties of doped carbon nanostructures [22–24], the adsorption of hydrogen sulfide, thiophene, and sulfur dioxide on the surfaces of simple and N-, P-, S-, Si-, Al-, and B-doped CNTs were studied. Consequently, in addition to the calculation of adsorption energies, molecular orbital properties, and optimized parameters, density functional theory (DFT) was employed to obtain the energy gaps ($E_g$) to examine the sensor abilities of doped nanotubes versus desired molecules. Finally, the interaction parameters were investigated using NBO calculations. The adsorption of hydrogen sulfur, sulfur dioxide, and thiophene on the surface of various doped CNTs was compared with simple CNTs to examine the effects of doping on them.

## 2. Materials and Methods

All the calculations were performed by Gaussian 09 program package [25] and using density functional theory (DFT) at $\omega$-B97X-D/6-31+g* level of theory. DFT calculations were used because they could reproduce exact energy values, comparative with the most expensive MP2 methods [26]. $\omega$-B97X-D functional is a DFT method based on long-range corrected hybrid density functions with consideration of empirical dispersion; its result accuracy and reproducibility has been validated through comparison with theoretical and experimental data [27]. This method is adequately modified for calculation of non-covalent interaction compared with standard DFT methods such as B3LYP, which made it desirable for calculation of CNTs and selected sulfur-based compounds interactions [28]. The open-shell calculations were carried out on the structures with the even electron number and the closed-shell calculations were employed on the other structures.

The optimization processes were carried out without any symmetric restriction. In addition, for the complexes of doped CNTs adsorbates, several systems starting from different relative positions and various conformations of the adsorbents were considered and, finally, the structure with the minimum energy value was selected for each case. The integral equation formalism variant of Tomasi's polarized continuum (IEFPCM) model [29] was employed using the SCRF keyword to calculate the free energy of solvation. Natural bond orbitals (NBO) calculations for all structures were performed by employing NBO 5.0 [17], as implemented in Gaussian. The adsorption energies for all interactions were obtained from Equation (1), by considering the basis set superposition error (BSSE) and thermodynamics correction.

$$\Delta E_{ad} = E_{complx} - (E_{adsorbent} + E_{adsorbate}) \tag{1}$$

Koopman's theorem was employed to calculate reactivity parameters for all structures. Consequently, global softness (S), chemical hardness ($\eta$), chemical potential ($\mu$), and electrophilicity index ($\omega$) were obtained using Equations (2)–(5).

$$M = (E\,(LUMO) + E\,(HOMO))/2 \tag{2}$$

$$H = (E\,(LUMO) - E\,(HOMO))/2 \tag{3}$$

$$S = 1/\eta \tag{4}$$

$$\omega = \mu^2/2\eta \tag{5}$$

## 3. Results and Discussions

### 3.1. Optimized Parameters

In this work, the computations were started from a simple (5,5)-carbon nanotube (simply named N) in which its ends were saturated with 10 hydrogen atoms. This model has been used in the previous studies of this group, since it was decided to use the same model for CNTs in all works for consistency and comparability of the results. Furthermore, this model is a common model in many other reports because the computational costs of the work are reduced without employing important approximation; a higher theoretical model could be employed if affordable in the future. Then, six doped structures containing one doped atom of aluminum (AN), boron (BN), nitrogen (NN), phosphorus (PN), sulfur (SN), and silicon (SiN) were made by the replacement of one carbon atom with the heteroatom. All heteroatoms were added in the same place, located in the middle of the CNTs to create uniform models for better comparison. Optimizations of these structures (as adsorbents), were performed at ω-B97X-D/6-31+G* level of theory. The optimized structures are shown in Figure 1 and they were used to extract the molecular parameters, as listed in Table 1.

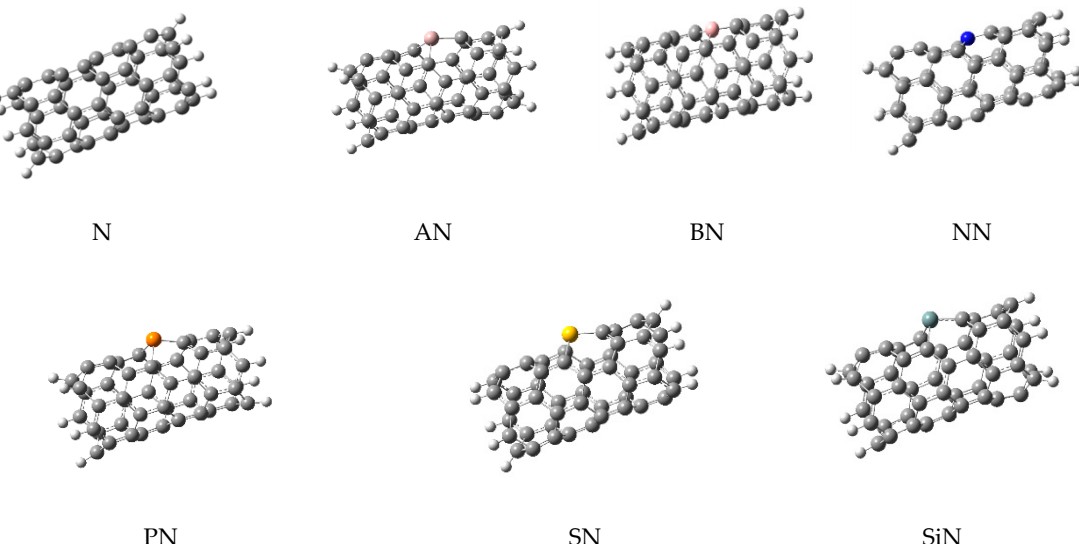

**Figure 1.** The optimized structures of simple and doped nanotubes employed in this work.

**Table 1.** The most important molecular parameters of simple and doped nanotubes, alone and in complex, with $H_2S$, $SO_2$, and thiophene. All distances were reported in Å.

| | N Alone | N-$H_2S$ | | N-$SO_2$ | | N-Thiophene | |
|---|---|---|---|---|---|---|---|
| | C-X (Av.) [a] | C-X (Av.) [a] | N-M [b] | C-X (Av.) [a] | N-M [b] | C-X (Av.) [a] | N-M [b] |
| N | 1.439 | 1.442 | 2.391 | 1.444 | 2.974 | 1.444 | 3.243 |
| AN | 1.913 | 1.933 | 2.330 | 1.977 | 1.789 | 1.935 | 2.307 |
| BN | 1.526 | 1.590 | 2.163 | 1.594 | 1.448 | 1.538 | 2.804 |
| NN | 1.441 | 1.445 | 2.284 | 1.447 | 2.979 | 1.436 | 3.162 |
| PN | 1.870 | 1.871 | 2.567 | 1.867 | 3.300 | 1.869 | 3.255 |
| SN | 1.863 | 1.863 | 2.174 | 1.868 | 2.018 | 1.860 | 3.134 |
| SiN | 1.868 | 1.871 | 2.291 | 1.872 | 2.930 | 1.865 | 3.067 |

[a] This distance shows the average values of three C-X bond lengths. [b] This parameter is related to the minimum distance between nanotubes (N) and small molecules (M).

As shown in Figure 1, the doping of nanotubes deformed their structure due to the difference between the atomic radius of the carbon and dopant atoms, which led to the differences in their bonds' lengths. Next, three sulfur-containing molecules ($H_2S$, $SO_2$, and thiophene, generally named M) were placed on the surface of each nanotube to obtain the complex structures. The next optimizations were performed on these complexes and their

important parameters are listed in Table 1. The optimized structures of these complexes are shown in Figure 2, illustrating the relative position of the adsorbents and adsorbates.

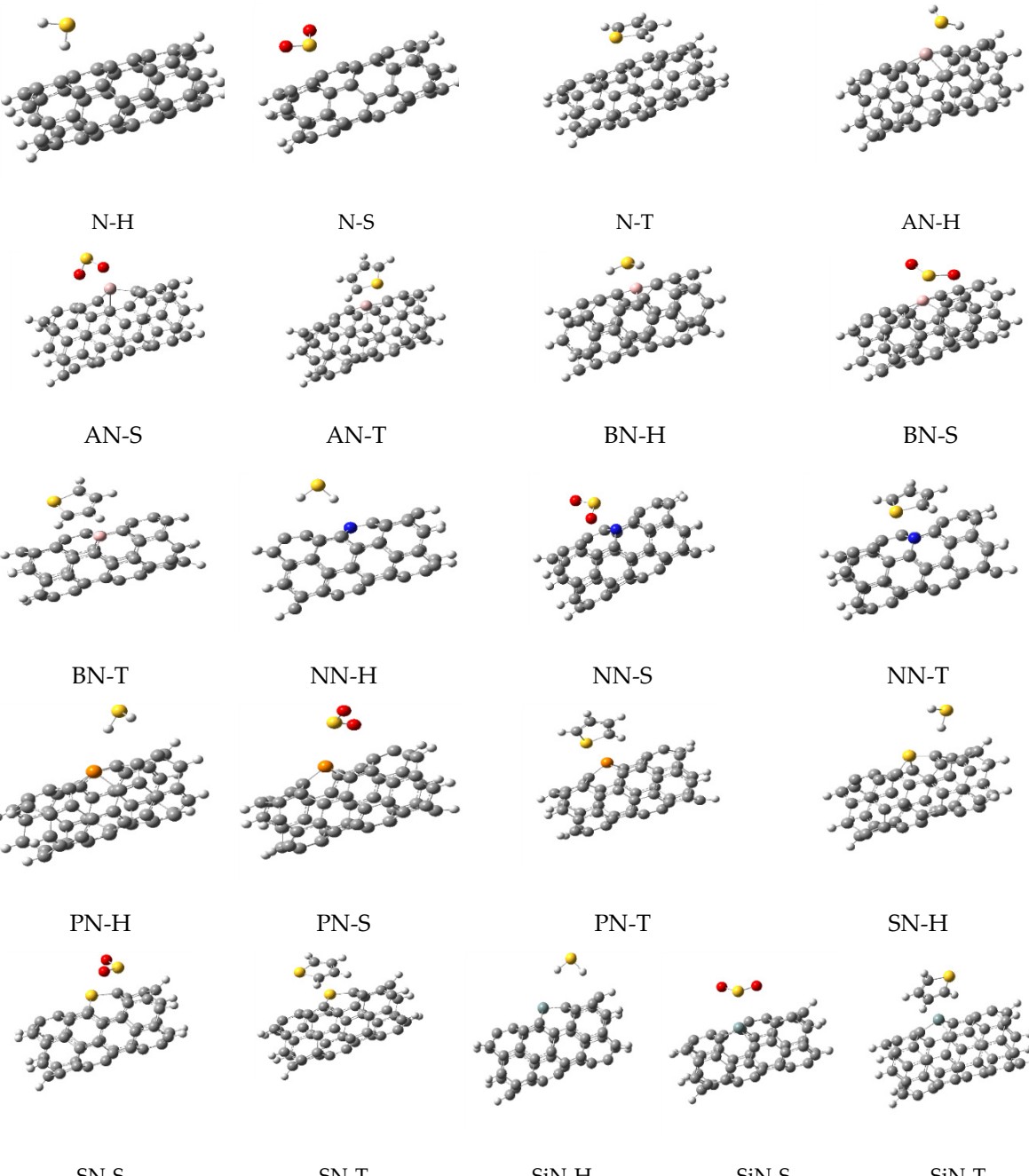

| N-H | N-S | N-T | AN-H |
| AN-S | AN-T | BN-H | BN-S |
| BN-T | NN-H | NN-S | NN-T |
| PN-H | PN-S | PN-T | SN-H |
| SN-S | SN-T | SiN-H | SiN-S | SiN-T |

**Figure 2.** The optimized structures for all complexes of nanotubes with $H_2S$, $SO_2$, and thiophene.

In Table 1, C-X values are related to the average bond lengths of dopant atoms (X) and their surrounding carbons (C). Comparing C-X bond lengths (X is the dopant atom) of each of the doped-CNTs with the similar value in their complexes (with $H_2S$, $SO_2$, and thiophene) showed almost similar values in all cases except in AN and BN. In these two structures, C-X values were increased dramatically in the complexes (versus the alone structure). The maximum variation in the C-X distances was observed in BN complexes where the values for BN, BN-$H_2S$, BN-$SO_2$, and BN-thiophene were 1.526, 1.590, 1.594, and 1.538 Å, respectively.

Averaging the N-M distances (between the nanotubes (N) and the small molecule (M)) among all studied complexes showed that the mean N-M distances for N-$H_2S$, N-$SO_2$, and N-thiophene were 2.314, 2.491, and 2.996 Å, respectively. Moreover, the average N-M distance for doped-CNTs were in this order: BN < AN < SN < SiN < NN < N < PN. Generally, the distance between adsorbent and adsorbate is affected by the strength of their interactions and the atomic sizes of adjacent atoms. The Al, P, Si, and S atoms are in the third row of the periodic table and have larger atomic radii. Therefore, expecting larger N-M values for these doped-CNTs is quite reasonable. Moreover, Al and B have fewer electronegativity values than the other doped heteroatoms, which led to the lower electron densities and stronger interactions with the electro-rich structures. Based on these arguments, it could be concluded that B and Al heteroatoms are the best candidates as dopant atoms for the adsorption of $H_2S$, $SO_2$, and thiophene, while PN and simple carbon nanotube are the worse adsorbents for these purposes.

### 3.2. Adsorption Energies

The adsorption energies for all complexes were calculated in the gas phase and water to study the strength of interactions, as shown in Table 2. According to the data listed in this table, all of the adsorption energies in the solvent have negative values, which indicate exothermic interactions of $H_2S$, $SO_2$, and thiophene with the studied CNTs. Comparing the adsorption energies in the gas phase and water indicated that the interactions of $H_2S$ and $SO_2$ with CNTs in the gas phase are more desirable than those in the solvent. In contrast, thiophene adsorption is quite better in the solvent rather than the gas phase. A comparison of all $E_{ad}$ values indicated that $SO_2$-containing CNTs have the most negative $E_{ad}$ values while $H_2S$-containing systems have the least negative ones. Finally, the average adsorption abilities of nanotubes based on $E_{ad}$ values were in this order: AN > BN > SiN > NN > PN > SN > N.

**Table 2.** Adsorption energies ($\Delta E_{ad}$), thermal correction of adsorption energies, basis set superposition error (BSSE) values, enthalpies of adsorptions ($\Delta H_{ad}$) and Gibbs free energy of adsorptions ($\Delta G_{ad}$) of all nanostructures with $H_2S$, $SO_2$, and thiophene in the gas (G) and water as solvent (W). All energy values are reported in kcal/mol by considering BSSE and thermodynamics correction.

| | N-$H_2S$ | | | N-$SO_2$ | | | N-Thiophene | | |
|---|---|---|---|---|---|---|---|---|---|
| | $\Delta E_{ad}$ (G) | $\Delta E_{ad}$ (W) | BSSE | $\Delta E_{ad}$ (G) | $\Delta E_{ad}$ (W) | BSSE | $\Delta E_{ad}$ (G) | $\Delta E_{ad}$ (W) | BSSE |
| N | −1.56 | −1.49 | 0.91 | −4.41 | −4.11 | 2.20 | −2.56 | −2.67 | 2.20 |
| AN | −5.15 | −5.04 | 1.89 | −7.08 | −6.87 | 2.29 | −5.69 | −6.02 | 2.43 |
| BN | −4.12 | −4.05 | 2.01 | −5.69 | −5.44 | 2.08 | −4.93 | −5.18 | 2.38 |
| NN | −3.21 | −3.08 | 1.01 | −3.67 | −3.53 | 2.45 | −3.51 | −3.64 | 2.40 |
| PN | −2.69 | −2.57 | 0.98 | −3.93 | −3.78 | 1.91 | −3.71 | −3.76 | 1.75 |
| SN | −2.99 | −2.92 | 1.08 | −4.14 | −3.85 | 1.97 | −2.72 | −2.95 | 2.04 |
| SiN | −2.46 | −2.47 | 0.96 | −5.17 | −4.95 | 2.90 | −3.60 | −3.74 | 2.26 |
| | $\Delta H_{ad}$ (G) | $\Delta H_{ad}$ (W) | Thermal correction | $\Delta H_{ad}$ (G) | $\Delta H_{ad}$ (W) | Thermal Correction | $\Delta H_{ad}$ (G) | $\Delta H_{ad}$ (W) | Thermal correction |
| N | −1.24 | −1.16 | 1.28 | −3.92 | −3.64 | 0.77 | −2.01 | −2.11 | 1.55 |
| AN | −4.82 | −4.71 | 1.66 | −6.62 | −6.41 | 0.98 | −5.11 | −5.44 | 1.84 |
| BN | −3.75 | −3.68 | 1.94 | −5.18 | −4.92 | 1.11 | −4.37 | −4.62 | 1.69 |
| NN | −2.87 | −2.73 | 0.60 | −3.22 | −3.08 | 0.96 | −2.97 | −3.09 | 1.76 |
| PN | −2.33 | −2.21 | 1.13 | −3.41 | −3.27 | 0.23 | −3.12 | −3.18 | 1.58 |
| SN | −2.70 | −2.62 | 1.35 | −3.66 | −3.38 | 0.85 | −2.22 | −2.46 | 1.42 |
| SiN | −2.11 | −2.13 | 1.47 | −4.74 | −4.51 | 0.81 | −3.06 | −3.21 | 1.55 |

**Table 2.** *Cont.*

|  | $\Delta G_{ad}$ (G) | $\Delta G_{ad}$ (W) | Thermal correction | $\Delta G_{ad}$ (G) | $\Delta G_{ad}$ (W) | Thermal Correction | $\Delta G_{ad}$ (G) | $\Delta G_{ad}$ (W) | Thermal correction |
|---|---|---|---|---|---|---|---|---|---|
| N | −0.13 | −0.06 | 2.39 | −2.40 | −2.13 | 2.99 | −0.36 | −0.47 | 3.20 |
| AN | −3.62 | −3.52 | 2.84 | −5.06 | −4.85 | 2.54 | −3.41 | −3.74 | 3.54 |
| BN | −2.43 | −2.35 | 3.26 | −3.57 | −3.32 | 2.72 | −2.70 | −2.94 | 3.36 |
| NN | −0.46 | −0.31 | 3.01 | −1.71 | −1.57 | 2.47 | −1.35 | −1.47 | 3.38 |
| PN | −1.19 | −1.08 | 2.27 | −1.86 | −1.71 | 1.78 | −1.48 | −1.55 | 3.22 |
| SN | −1.42 | −1.35 | 2.63 | −2.06 | −1.78 | 2.45 | −0.61 | −0.85 | 3.03 |
| SiN | −0.92 | −0.95 | 2.66 | −3.16 | −2.93 | 2.39 | −1.40 | −1.54 | 3.21 |

The order of adsorption energy values is similar to the previous section, and analyzing the $E_{ad}$ values confirmed that Al- and B-doped CNTs have the strongest interactions with sulfur-containing molecules. In contrast, CNTs and sulfur-doped CNTs are the worst adsorbent for these adsorbates.

*3.3. NBO Calculations*

In this study, the NBO program was used to calculate the partial atomic charges and the second-order perturbation energies, which can provide more details about CNT and sulfur-containing compounds interactions. The NBO atomic charges are listed in Table 3. In this table, X refers to the partial atomic charges of doped heteroatoms and the partial charge of nearest carbon atom refers to the adsorbate for simple carbon nanotubes. C (Av) refers to the average values for the atomic charges of carbon atoms connected to the heteroatoms. Moreover, the sulfur charges were used to report the atomic charges of the sulfur atom in $H_2S$, $SO_2$, and thiophene. Finally, the average charges of two adjacent hydrogen atoms in $H_2S$, two adjacent oxygen atoms in $SO_2$, and C1 and C5 in thiophene, were labeled as Y charges.

**Table 3.** NBO atomic charges (in atomic units) for all adsorbents, adsorbates, and complexes.

| Nanotube Charges | N Alone | | N-$H_2S$ | | N-$SO_2$ | | N-Thiophene | |
|---|---|---|---|---|---|---|---|---|
|  | C (Av) [a] | X [b] | C (Av) [a] | X [b] | C (Av) [a] | X [b] | C (Av) [a] | X [b] |
| N | 0.000 | 0.000 | −0.022 | −0.068 | −0.017 | −0.091 | −0.003 | −0.011 |
| AN | −0.498 | 1.673 | −0.493 | 1.568 | −0.445 | 1.750 | −0.536 | 1.747 |
| BN | −0.310 | 0.639 | −0.256 | 0.450 | −0.178 | 0.222 | −0.315 | 0.730 |
| NN | 0.219 | −0.381 | 0.223 | −0.414 | 0.205 | −0.422 | 0.231 | −0.381 |
| PN | −0.271 | 0.928 | −0.266 | 0.928 | −0.275 | 0.894 | −0.270 | 0.942 |
| SN | −0.195 | 0.844 | −0.223 | 0.851 | −0.233 | 0.953 | −0.194 | 0.856 |
| SiN | −0.391 | 1.170 | −0.391 | 1.146 | −0.386 | 1.157 | −0.398 | 1.203 |

| Adsorbate charges | N-$H_2S$ | | N-$SO_2$ | | N-thiophene | |
|---|---|---|---|---|---|---|
|  | S | Y (Av) [d] | S | Y (Av) [d] | S | Y (Av) [d] |
| small molecule [c] | −0.340 | 0.170 | 1.286 | −0.643 | 0.358 | −0.416 |
| N | −0.333 | 0.164 | 1.296 | −0.673 | 0.360 | −0.413 |
| AN | −0.242 | 0.235 | 0.967 | −0.827 | 0.494 | −0.507 |
| BN | −0.034 | 0.226 | 1.074 | −0.724 | 0.372 | −0.405 |
| NN | −0.356 | 0.172 | 1.262 | −0.657 | 0.364 | −0.415 |
| PN | −0.354 | 0.176 | 1.303 | −0.660 | 0.352 | −0.415 |
| SN | −0.361 | 0.169 | 1.289 | −0.888 | 0.367 | −0.416 |
| SiN | −0.340 | 0.165 | 1.216 | −0.725 | 0.364 | −0.415 |

[a] This value is the average of atomic charges of three carbon atoms connected to the doped heteroatom. [b] The charge of heteroatom in nanotubes. For N, this is the charge of the carbon atom nearest to the adsorbate. [c] Small molecule is implicated to $H_2S$, $SO_2$, and thiophene. [d] Y is the average of atomic charges of hydrogens in $H_2S$, oxygen atoms in $SO_2$, and C1 and C5 in thiophene.

In all complexes, the charge transfer process could be easily traced by the measurement of charge alteration of adsorbent or adsorbate versus their initial charges. Regarding

this, measurement of the charge transfers for $H_2S$ complexes revealed that BN and AN complexes have the maximum charge transfer values and the obtained values were in this order: BN > AN > NN > SiN > SN > PN. Furthermore, the maximum charge transfer for $SO_2$ complexes was related to BN and SN and charge transfer values were in this order: BN > SN > AN > NN > PN > SiN. Moreover, in the thiophene complex, the maximum charge transfer belonged to BN and AN and they were in this order: BN > AN > SiN > PN > SN > NN. Therefore, it could be concluded that NBO charge values have strong agreement with the results of previous sections on the positive effect of B and Al as doped atoms on the improvement of the studied adsorption processes.

In addition to the atomic charges, NBO calculation was used to investigate the E2 values in all 21 complexes (Table 4). E2 is the second-order perturbation energies for donor-acceptor interactions and it shows the strength of the donor-acceptor interaction. The sum of these second-order perturbation energy values is reported in the last column. According to these values, the highest second-order perturbation energies belonged to BN and AN complexes. The results of the NBO calculations confirmed the results of previous sections and introduced BN and AN as the best adsorbents.

**Table 4.** The strongest second-order perturbation energies (E2) (in kcal/mol) for the donor-acceptor transaction for all complexes.

| Complex | Donor | Acceptor | E2 | Donor | Acceptor | E2 | Donor | Acceptor | E2 | Sum [a] |
|---|---|---|---|---|---|---|---|---|---|---|
| N-H | LPC | σ*S-H | 0.55 | σS-H | LPC | 0.36 | σC-C | σ*S-H | 0.17 | 1.08 |
| N-S | LPC | Π*S-O | 1.65 | Π*c-c | σ*S-O | 0.83 | LPC | σ*S-O | 0.61 | 3.09 |
| N-T | Π*c-c | Π*c-c | 0.60 | Π*c-c | Π*c-c | 0.40 | Π*c-c | Π*c-c | 0.17 | 1.17 |
| AN-H | LPS | LP*Al | 1.62 | LPS | LP*Al | 1.34 | LPS | LP*Al | 0.61 | 3.57 |
| AN-S | LPO | LP*Al | 3.09 | LP*Al | σ*S-O | 1.39 | LPO | LP*Al | 1.09 | 5.57 |
| AN-T | Πc-c | LP*Al | 1.40 | Πc-c | LP*Al | 0.95 | Π*c-c | LP*Al | 0.89 | 3.24 |
| BN-H | LPS | LP*B | 1.32 | LPS | LP*B | 0.95 | CRS | LP*B | 0.61 | 2.88 |
| BN-S | LPO | σ*C-C | 1.75 | LPO | σ*C-C | 1.36 | LPO | RY*C | 0.56 | 3.67 |
| BN-T | Π*C-C | LP*B | 1.35 | ΠC-C | LP*B | 1.32 | Π*C-C | LP*B | 0.65 | 3.32 |
| NN-H | LPN | σ*S-H | 0.71 | ΠC-C | σ*S-H | 0.64 | σS-H | RY*C | 0.19 | 1.54 |
| NN-S | LPN | LP*S | 1.04 | ΠC-C | LP*S | 0.58 | σS-O | RY*C | 0.24 | 1.86 |
| NN-T | Π*C-C | Π*C-C | 0.70 | ΠC-C | Π*C-C | 0.57 | Π*C-C | Π*C-C | 0.42 | 1.69 |
| PN-H | ΠC-C | σ*S-H | 0.68 | Π*C-C | σ*S-H | 0.31 | Π*C-C | σ*S-H | 0.29 | 1.28 |
| PN-S | LPP | LP*S | 1.22 | ΠC-C | LP*S | 0.92 | LP*S | σ*C-P | 0.37 | 2.51 |
| PN-T | Π*C-C | Π*C-C | 1.13 | LPS | σ*C-P | 0.67 | ΠC-C | Π*C-C | 0.45 | 2.25 |
| SN-H | ΠC-C | σ*S-H | 0.84 | Π*C-C | σ*S-H | 0.36 | σS-H | Π*C-C | 0.24 | 1.44 |
| SN-S | LPO | σ*C-S | 1.56 | LPO | σ*C-S | 1.06 | LPO | σ*C-C | 0.32 | 2.94 |
| SN-T | ΠC-C | σ*C-S | 0.56 | Π*C-C | Π*C-C | 0.34 | Π*C-C | Π*C-C | 0.23 | 1.13 |
| SiN-H | σC-Si | σ*S-H | 0.57 | ΠC-C | σ*S-H | 0.38 | Π*C-C | σ*S-H | 0.20 | 1.15 |
| SiN-S | LPS | σ*C-Si | 1.82 | LPS | σ*C-Si | 1.02 | LPS | LP*Si | 0.81 | 3.65 |
| SiN-T | Π*C-C | Π*C-C | 1.19 | Π*C-C | Π*C-C | 0.42 | Π*C-C | LP*Si | 0.28 | 1.89 |

[a] This is the sum of three E2 values listed in the same row.

### 3.4. Reactivity Parameters

In the final part of this study, the molecular orbital population analyses were employed to obtain the HOMO-LUMO band gaps and reactivity parameters for all adsorbents and complexes. The results of these calculations are listed in Table 5.

The HOMO-LUMO energy gaps for various nanotubes were in this order: AN (0.197 eV) > PN (0.184 eV) > NN (0.178 eV) > SN (0.166 eV) > N (0.161 eV) > BN (0.155 eV) > SiN (0.150 eV). It seems that the conductivity of carbon nanotubes was enhanced by doping with B and Si heteroatoms. In addition, according to the energy gap values for studied complexes, the intramolecular interactions between adsorbents and adsorbates changed the bandgap for each complex. Therefore, the HOMO-LUMO band gaps of CNTs could be changed in the presence of $H_2S$, $SO_2$, and thiophene. It should be mentioned that by adsorption of $SO_2$, the $E_g$ values of CNTs were reduced more, compared with $H_2S$ and thiophene, which can be interpreted as the stronger interactions of $SO_2$ with the studied carbon

nanotube. The doping of Al, B, N, and P atoms could affect E (HOMO) and E (LUMO) values, while for S- and Si-doped nanotubes the E (HOMO) values were exactly equal to the simple carbon nanotube. A comparison of chemical potential ($\mu$) values indicated that the highest chemical potential was related to BN and the order of these values is: BN > SN > AN, N > NN > PN > SiN. Moreover, a slight decrease was observed in chemical potential ($\mu$) values of the complexes. Among all complexes, $SO_2$-containing systems showed the most decline in chemical potential values versus their related simple and doped-CNTs. The global softness (S) values and chemical hardness ($\eta$) is related to $E_g$ values and no further explanation is required. Finally, the electrophilicity index ($\omega$) for various nanotubes was in this order: SiN > N > SN > NN > PN > BN > AN. It could be concluded that electrophilicity indices were not meaningfully affected by doping nanotubes. Finally, a comparison of electrophilicity index in complexes indicated that in all complexes SO2 and $H_2S$ could increase the electrophilicity indices, while thiophene decreased them. This section should provide a concise and precise description of the experimental results, their interpretation, as well as the experimental conclusions that can be drawn.

**Table 5.** Energies of HOMO and LUMO levels, energy gaps ($E_g$), chemical potential ($\mu$), chemical hardness ($\eta$), global softness (S), and electrophilicity index ($\omega$) for all structures (all energy values in eV).

| Complex | E (HOMO) | E (LUMO) | $E_g$ | $\mu$ | $\eta$ | S | $\omega$ |
|---------|----------|----------|-------|-------|--------|------|----------|
| N | −0.234 | −0.073 | 0.161 | −0.154 | 0.080 | 12.454 | 0.147 |
| N-H | −0.237 | −0.077 | 0.160 | −0.157 | 0.080 | 12.508 | 0.155 |
| N-S | −0.242 | −0.093 | 0.149 | −0.168 | 0.074 | 13.426 | 0.189 |
| N-T | −0.233 | −0.072 | 0.160 | −0.152 | 0.080 | 12.467 | 0.145 |
| AN | −0.252 | −0.055 | 0.197 | −0.154 | 0.099 | 10.149 | 0.120 |
| AN-H | −0.245 | −0.053 | 0.192 | −0.149 | 0.096 | 10.440 | 0.116 |
| AN-S | −0.249 | −0.079 | 0.169 | −0.164 | 0.085 | 11.801 | 0.159 |
| AN-T | −0.240 | −0.040 | 0.200 | −0.140 | 0.100 | 9.994 | 0.097 |
| BN | −0.217 | −0.061 | 0.155 | −0.139 | 0.078 | 12.870 | 0.125 |
| BN-H | −0.240 | −0.048 | 0.192 | −0.144 | 0.096 | 10.440 | 0.108 |
| BN-S | −0.237 | −0.075 | 0.162 | −0.156 | 0.081 | 12.318 | 0.149 |
| BN-T | −0.232 | −0.059 | 0.174 | −0.145 | 0.087 | 11.527 | 0.122 |
| NN | −0.244 | −0.067 | 0.178 | −0.155 | 0.089 | 11.268 | 0.136 |
| NN-H | −0.251 | −0.069 | 0.182 | −0.160 | 0.091 | 10.984 | 0.140 |
| NN-S | −0.250 | −0.079 | 0.171 | −0.165 | 0.086 | 11.667 | 0.158 |
| NN-T | −0.242 | −0.062 | 0.180 | −0.152 | 0.090 | 11.083 | 0.128 |
| PN | −0.248 | −0.063 | 0.184 | −0.156 | 0.092 | 10.854 | 0.131 |
| PN-H | −0.248 | −0.068 | 0.180 | −0.158 | 0.090 | 11.131 | 0.139 |
| PN-S | −0.252 | −0.097 | 0.155 | −0.174 | 0.078 | 12.902 | 0.196 |
| PN-T | −0.245 | −0.056 | 0.189 | −0.150 | 0.094 | 10.590 | 0.120 |
| SN | −0.234 | −0.068 | 0.166 | −0.151 | 0.083 | 12.056 | 0.138 |
| SN-H | −0.239 | −0.075 | 0.164 | −0.157 | 0.082 | 12.192 | 0.150 |
| SN-S | −0.250 | −0.099 | 0.152 | −0.174 | 0.076 | 13.165 | 0.200 |
| SN-T | −0.232 | −0.067 | 0.165 | −0.149 | 0.083 | 12.104 | 0.135 |
| SiN | −0.234 | −0.083 | 0.150 | −0.158 | 0.075 | 13.293 | 0.167 |
| SiN-H | −0.241 | −0.092 | 0.149 | −0.166 | 0.075 | 13.405 | 0.185 |
| SiN-S | −0.245 | −0.100 | 0.145 | −0.172 | 0.073 | 13.759 | 0.204 |
| SiN-T | −0.232 | −0.082 | 0.151 | −0.157 | 0.075 | 13.280 | 0.164 |

## 4. Conclusions

In this study, the sensor abilities and adsorption potentials of simple and Al-, B-, N-, S-, P-, and Si-doped CNTs interacting with some sulfur-containing molecules ($H_2S$, $SO_2$, and thiophene) were investigated theoretically. In this line, DFT calculations were used to calculate the adsorption energies and their related parameters. The results showed an exothermic interaction of $H_2S$, $SO_2$, and thiophene with CNT and doped CNTs. The maximum negative adsorption energies belonged to AN and BN. The NBO program was used to calculate second-order perturbation energies related to the interactions between

adsorbents and adsorbates. The highest perturbation energies were related to BN and AN. Finally, population analyses were performed to calculate the HOMO-LUMO energy gaps and reactivity parameters.

Comparing the results of this work with the previous studies showed similar findings in all of these works. For example, the theoretical studies of Sonawane et al. on the adsorption of $SO_2$ on silicon-doped CNTs showed that the presence of silicon has a meaningful effect on the adsorption energies [30]. Moreover, the work of Sun et al. showed the doping of carbon-based materials with nitrogen could enhance the effective surface area for the adsorption of $SO_2$ [31]. In this line, there are a number of studies showing the enhancement effects of doping carbon materials on their adsorption potencies for the studied molecules [32].

Briefly, the results demonstrated the favorability of adsorption of $SO_2$ by CNTs (BN and AN, in particular). Furthermore, the observed changes in the energy gap values of the BN and AN complexes (versus the CNTs alone) introduced Al- and B-doped CNTs as excellent candidates for employment in sensor devices.

**Author Contributions:** Conceptualization, H.T.; methodology, H.H.; software, H.H. and H.T.; formal analysis, H.H.; investigation H.H. and H.T.; resources, H.T.; data curation, H.H.; writing—original draft preparation, H.H.; writing—review and editing, H.T.; supervision, H.T.; project administration, H.T. All authors have read and agreed to the published version of the manuscript.

**Funding:** This research received no external funding.

**Acknowledgments:** We are thankful to the National High-Performance Computing Center (NHPCC) at Isfahan University of Technology (http://nhpcc.iut.ac.ir, accessed on 1 July 2021) for providing computational facilities (Rakhsh supercomputer) for this study.

**Conflicts of Interest:** The authors declare no conflict of interest. The funders had no role in the design of the study; in the collection, analyses, or interpretation of data; in the writing of the manuscript, or in the decision to publish the results.

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
