# Peer review of "A DFT Study on the Interaction of Doped Carbon Nanotubes with H2S, SO2 and Thiophene"

_quantumrep, doi:10.3390/quantum3030023_

Round 1
Reviewer 1 Report
In this manuscript, the authors performed ab initio calculations to study the interactions between three different S-containing molecules, i.e., H2S, SO2, thiophene, and clean/doped carbon nanotubes. The authors presented adsorption energies, NBO parameters, energy gaps and reactivity parameters.
The computational methods employed by the authors are appropriate. Moreover, the data presented in this manuscript will contribute to the computational community and will have a broad audience.
- There are capitalizations at unusual places in the manuscript, please check with the standard notations in literature, and address this issue.
- The authors need to give more details about the locations of heteroatom in the structure. More specifically, dopants near the edge may not be electronically equivalent to that in the middle section.
- The sentences in lines 127-131 are confusing. Please clarify and revise the manuscript if necessary.
- In Lines 156-158, the authors compared adsorption energies in gas phase with those in solvents. What is the solvation model? This information is not given.
- Re-plot all DOS plots in Figure 3, the legends and axes are not readable.
- The electronic structure calculations, including energy gaps of CNTs are common. Need to compare and cite energy gaps values if possible.
This manuscript is probably publishable, but needs revision before acceptance.
Author Response
Dear referee
Thanks for the comments on our manuscript (Manuscript ID: quantumrep-1242622). Following, you may find our responses to the points requested more explanations/corrections. We hope the present format of the manuscript will be suitable for publishing.
Best Regards
Hossein Tavakol
Comments and Suggestions for Authors
In this manuscript, the authors performed ab initio calculations to study the interactions between three different S-containing molecules, i.e., H2S, SO2, thiophene, and clean/doped carbon nanotubes. The authors presented adsorption energies, NBO parameters, energy gaps and reactivity parameters. The computational methods employed by the authors are appropriate. Moreover, the data presented in this manuscript will contribute to the computational community and will have a broad audience.
- There are capitalizations at unusual places in the manuscript, please check with the standard notations in literature, and address this issue.
Response: Sorry, all of these mistakes (unusual capitalization) were corrected.
- The authors need to give more details about the locations of heteroatom in the structure. More specifically, dopants near the edge may not be electronically equivalent to that in the middle section.
Response: More details about the locations of heteroatom in the structure were added (section 3.1., lines 10-11).
- The sentences in lines 127-131 are confusing. Please clarify and revise the manuscript if necessary.
Response: The whole of them were revised to be more clarified.
- In Lines 156-158, the authors compared adsorption energies in gas phase with those in solvents. What is the solvation model? This information is not given.
Response: The details of the solvation model were presented in the second paragraph of section 2 (materials and methods).
- Re-plot all DOS plots in Figure 3, the legends and axes are not readable.
Response: All DOS plots were deleted because of the request of the second referee.
- The electronic structure calculations, including energy gaps of CNTs are common. Need to compare and cite energy gaps values if possible.
Response: Energy gaps could be found in the fourth column of Table 5 and the related discussions were mentioned after that table.
This manuscript is probably publishable, but needs revision before acceptance.
Reviewer 2 Report
This work is devoted to the study of the adsorption of sulfur-containing small molecules (H2S, SO2, thiophene) on simple and Al, B, N, S, P, Si-doped carbon nanotubes. The main result of this work is the illustration of the highest H2S/SO2/thiophene adsorption activity of Al- and B-doped nanotubes. In addition, several attempts have been made to analyze the nature of the corresponding non-covalent interactions.
Unfortunately, the manuscript has serious flaws:
- А key part of adsorption analysis is the optimization of the geometry of molecule-adsorbate systems starting from different relative positions (see e.g. DOI 10.1007/s10450-020-00218-3). As a result, the one obtains various binding geometries and the corresponding interaction energies. This works lacks this step, as a result, only one geometry is available for each system, raising questions about the correctness of the dependence of the adsorption energies on the dopant.
- Tables 3 and 4, containing info on NBO charges and NBO perturbation energies, are essentially raw data. Indeed, an analysis of non-covalent interactions is usually based on a change of atomic charges, and the strongest interaction of natural orbitals in isolation from the rest is meaningless. At the same time, standard tools for the analysis of non-covalent interactions, such as maps of deformational electron density and electrostatic potential maps, were not used in this work.
- Usage of DOS for molecular systems is extremely surprising and produces a false sense that periodic DFT calculations were carried out. Considering that DOS is a concept of solid-state theory, the description of a method of transforming molecular orbitals energies into DOS of periodic structure supported by relevant references is required.
In addition, the article is inconvenient to read due to several technical issues, e.g. some figures describing adsorption geometry are ambiguous, and there are no XYZ-files in the supporting information.
To summarize: the used method of searching for nanotube-molecule complexes does not allow one to conclude that the obtained series of adsorption energies is correct even on a qualitative level. The performed quantum chemical analysis of the considered systems has not been analyzed in any way, only raw results are presented and discussed poorly with minimal attempts to analyze the nature of non-covalent bonding. The most common tools for analysis of the nature of intermolecular interactions were not used (maps of deformational electron density, electrostatic potential maps, and detailed (!) NBO analysis). Thus, the present work can be viewed as a rough draft for a future article, but not as a finished publication. As a result, I cannot recommend a work for publication in the journal Quantum Reports or other theoretical and chemical journals of the MDPI publisher (e.g., Molecules).
Author Response
Dear referee
Thanks for the comments on our manuscript (Manuscript ID: quantumrep-1242622). Following, you may find our responses to the points requested more explanations/corrections. We hope the present format of the manuscript will be suitable for publishing.
Best Regards
Hossein Tavakol
Comments and Suggestions for Authors
This work is devoted to the study of the adsorption of sulfur-containing small molecules (H2S, SO2, thiophene) on simple and Al, B, N, S, P, Si-doped carbon nanotubes. The main result of this work is the illustration of the highest H2S/SO2/thiophene adsorption activity of Al- and B-doped nanotubes. In addition, several attempts have been made to analyze the nature of the corresponding non-covalent interactions.
Unfortunately, the manuscript has serious flaws:
- А key part of adsorption analysis is the optimization of the geometry of molecule-adsorbate systems starting from different relative positions (see e.g. DOI 10.1007/s10450-020-00218-3). As a result, the one obtains various binding geometries and the corresponding interaction energies. This works lacks this step, as a result, only one geometry is available for each system, raising questions about the correctness of the dependence of the adsorption energies on the dopant.
Response: As you mentioned, surely this is the standard procedure for this type of studies and we have done the same. Moreover, various conformations of the adsorbates have been considered and finally, the structure with the minimum energy value was selected for each case. However, we normally don’t mention these explanations. Therefore, we added these explanations were added to the manuscript (the second paragraph of section 2) to clarify this fact.
- Tables 3 and 4, containing info on NBO charges and NBO perturbation energies, are essentially raw data. Indeed, an analysis of non-covalent interactions is usually based on a change of atomic charges, and the strongest interaction of natural orbitals in isolation from the rest is meaningless. At the same time, standard tools for the analysis of non-covalent interactions, such as maps of deformational electron density and electrostatic potential maps, were not used in this work.
Response: There are several references related to use the same data for the analysis of non-covalent interactions and we have severally used them in the all of our previous related work.
- Usage of DOS for molecular systems is extremely surprising and produces a false sense that periodic DFT calculations were carried out. Considering that DOS is a concept of solid-state theory, the description of a method of transforming molecular orbitals energies into DOS of periodic structure supported by relevant references is required.
Response: Surely your statement is correct and we have not found the requested reference and the relation between these diagrams and the work is poor. Therefore, they were deleted from the manuscript and SI.
Reviewer 3 Report
Manuscript ID quantumrep-1242622
----------------------------------
The manuscript contains the theoretical study of physisorption of the sulfur-containing model systems representing environmental pollutants, namely H2S, SO2, and thiophene on the pristine carbon nanotubes and their analogs doped with several heteroatoms. The topic can be interesting for a wide audience due to the continuous search of adsorbents for efficient air and water treatment. The nanotubes are considered as one of the most promising materials for these types of applications, therefore numerous literature reports (both theoretical and experimental) are available.
The presented manuscript applies several tools of computational
chemistry such as geometry optimization, supermolecular interaction energy estimation, partial charge analysis, and density-of-states-study. The
selection of the tools as well as the selection of the approach (namely
omega-B97X-D/6-31G+(D) in a vacuum and in PCM/water) seems to be reliable, however, certain shortcomings can be improved:
1. line 28.: "several studies (...) have been performed to evaluate the
absorption properties of CNTs and their doped derivatives" -- word
"several" seems to be inadequate in this context, since the Web of
Knowledge gives 24494 hits for "carbon nanotube" and "adsorption" in the topic (with 900 reviews) while adding "doped" gives 2932 results. Thus, it is indeed more than 'several'. Therefore, the citations should be supplemented with the corresponding references from the high-impact journals.
2. The considered systems are either closed-shell (and thus computationally
simple) or open-shell, with an even number of electrons (namely N, B, P,
Al-doped tubes) and no explanation is given on the way of the open-shell
system treatment. This methodological explanation should be provided.
3. DFT approach is applied for the HOMO-LUMO band gap estimation. However it is known from the fundamental work of Perdew (IJQC 1985, 28, 497-523, "Density functional theory and the bandgap problem") that DFT
significantly underestimates (by 30-50%) the bandgap due to the
self-interaction problem. How much this drawback of the methodology affects the results presented in the current contribution? Is the qualitative tendency presented in Tab. 5 and figures of DOS reproduced also with the
conventional wave-function theory? This question seems to be additionally
justified in light of the relatively small energy differences between
the investigated systems.
4. The discussion of the geometrical parameters taking the average of the
several adjacent bonds is hard to follow. Additionally, the only intermolecular geometrical parameter presented for the adsorbent-adsorbate complexes is the minimal (atom-atom?) distance between the subsystems. However, taking into account the allowed free motion of the weakly interactiong pollutant on the surface of the tube and no dynamical sampling of the system, this parameter seems to be vague and not representing any data of statistical importance. This is additionally important due to the single configuration of the small molecule on the tube surface and no averaging over the possible (even selected) structures. Was the geometry optimization for the complexes performed always for the same initial configuration? The different starting point can strongly affect the obtained complex structure and no technical details are given throughout the manuscript on the statistical significance. This issue should be elaborated (additional test calculations for instance) and the comparison of the obtained complexes for the different tubes should be performed for instance via the imposition of the corresponding structures to visualize the differences and similarities. The structures presented in Fig. 2 are really difficult to analyze since each and every tube is presented from a different orientation and no comparison is possible with these 2D structures upon different doping atoms and adsorbates.
5. The continuum solvation model is applied to describe the influence of the solvent presence on the absorption energy. However, both (some of) the nanotubes and the adsorbates exhibit the possibility of hydrogen bond formation, which can strongly affect the adsorption energy. Thus, the introduction of water, which in general can compete for the hydrogen bonds with the adsorbates or facilitate the hydrogen bonding networks with the adsorbate far from the surface, can influence the stability of the complexes. Again - what is the statistical importance of the continuum solvation calculations, where no hydrogen bond is possible and what justifies this choice of approach (despite its simplicity)?
Additionally, several issues must be improved including thorough language
editing, subscripts in SO2 and H2S, 'omega' instead of 'w', and capital
letters in the name of omega-B97X-D functional (line 71 -- also the basis set symbol should be unified with line 108), E_ad in line 160-162 with the subscript. Also, the bibliography needs to be corrected (for instance ref. 24 and 25 involves Structural Chemistry rather than Journal of Structural Chemistry and in ref. 24 the year, volume, and pages should be corrected).
Therefore, I would recommend reconsidering the manuscript for publication after major revision.
Author Response
Dear referee
Thanks for the comments on our manuscript (Manuscript ID: quantumrep-1242622). Following, you may find our responses to the points requested more explanations/corrections. We hope the present format of the manuscript will be suitable for publishing.
Best Regards
Hossein Tavakol
Comments and Suggestions for Authors
The manuscript contains the theoretical study of physisorption of the sulfur-containing model systems representing environmental pollutants, namely H2S, SO2, and thiophene on the pristine carbon nanotubes and their analogs doped with several heteroatoms. The topic can be interesting for a wide audience due to the continuous search of adsorbents for efficient air and water treatment. The nanotubes are considered as one of the most promising materials for these types of applications, therefore numerous literature reports (both theoretical and experimental) are available.
The presented manuscript applies several tools of computational chemistry such as geometry optimization, supermolecular interaction energy estimation, partial charge analysis, and density-f-states-study. The selection of the tools as well as the selection of the approach (namely
omega-B97X-D/6-31G+(D) in a vacuum and in PCM/water) seems to be reliable, however, certain shortcomings can be improved:
- line 28.: "several studies (...) have been performed to evaluate the absorption properties of CNTs and their doped derivatives" – word "several" seems to be inadequate in this context, since the Web of Knowledge gives 24494 hits for "carbon nanotube" and "adsorption" in the topic (with 900 reviews) while adding "doped" gives 2932 results. Thus, it is indeed more than 'several'. Therefore, the citations should be supplemented with the corresponding references from the high-impact journals.
Response: OK, the mentioned word was changed (to a more appropriate phrase) to emphasis on the huge number of the related works. Moreover, since we could not mention all these references, the appropriate references (from the high-impact journals) were inserted, instead of less important ones.
- The considered systems are either closed-shell (and thus computationally simple) or open-shell, with an even number of electrons (namely N, B, P, Al-doped tubes) and no explanation is given on the way of the open-shell system treatment. This methodological explanation should be provided.
Response: The open-shell was considered when necessary (systems with the even number of electron). The related explanations were added to the method section to clarify the use of these systems.
- DFT approach is applied for the HOMO-LUMO band gap estimation. However, it is known from the fundamental work of Perdew (IJQC 1985, 28, 497-523, "Density functional theory and the bandgap problem") that DFT significantly underestimates (by 30-50%) the bandgap due to the
self-interaction problem. How much this drawback of the methodology affects the results presented in the current contribution? Is the qualitative tendency presented in Tab. 5 and figures of DOS reproduced also with the conventional wave-function theory? This question seems to be additionally justified in light of the relatively small energy differences between the investigated systems.
Response: Surely the above comment is correct and the HOMO-LUMO band gap in the DFT calculation is a raw estimate and the results of DOS is not so useful. In fact, another referee has the same opinion about the DOS plots. Therefore, we deleted all DOS plots from the manuscript because we have not reliable reason to keep them. However, for the HOMO-LUMO band gap, despite the high-estimation of them in the quantitative reports, they have been used in many reports for the qualitative and comparative reporting band gaps. Indeed, since we have compared the different band gaps of the related structures, these estimations seem more reliable. We have found lots of similar data, in the related reports that used DFT calculations to report the band gaps of CNTs; because the use of the more reliable calculations (such as MP2, …) for such large systems are impossible.
- The discussion of the geometrical parameters taking the average of the several adjacent bonds is hard to follow. Additionally, the only intermolecular geometrical parameter presented for the adsorbent-adsorbate complexes is the minimal (atom-atom?) distance between the subsystems. However, taking into account the allowed free motion of the weakly interactiong pollutant on the surface of the tube and no dynamical sampling of the system, this parameter seems to be vague and not representing any data of statistical importance. This is additionally important due to the single configuration of the small molecule on the tube surface and no averaging over the possible (even selected) structures. Was the geometry optimization for the complexes performed always for the same initial configuration? The different starting point can strongly affect the obtained complex structure and no technical details are given throughout the manuscript on the statistical significance. This issue should be elaborated (additional test calculations for instance) and the comparison of the obtained complexes for the different tubes should be performed for instance via the imposition of the corresponding structures to visualize the differences and similarities. The structures presented in Fig. 2 are really difficult to analyze since each and every tube is presented from a different orientation and no comparison is possible with these 2D structures upon different doping atoms and adsorbates.
Response: For more clarification, this is the standard procedure for this type of studies to perform optimization of the geometries of molecule-adsorbate systems starting from different relative positions and we have done the same. Moreover, various conformations of the adsorbates have been considered and finally, the structure with the minimum energy value was selected for each case. However, we normally don’t mention these explanations. Therefore, we added these explanations were added to the manuscript (the second paragraph of section 2) to clarify this fact.
In Figure 2, we have tried to show each structure, from the best view, in which everyone could see all of the important structural aspects of the structures. In fact, because of the different structures of the adsorbate and different configurations for the adsorbent-adsorbate complexes, showing all of them from a similar view was impossible, if we want to show all important parts of these structures.
- The continuum solvation model is applied to describe the influence of the solvent presence on the absorption energy. However, both (some of) the nanotubes and the adsorbates exhibit the possibility of hydrogen bond formation, which can strongly affect the adsorption energy. Thus, the introduction of water, which in general can compete for the hydrogen bonds with the adsorbates or facilitate the hydrogen bonding networks with the adsorbate far from the surface, can influence the stability of the complexes. Again - what is the statistical importance of the continuum solvation calculations, where no hydrogen bond is possible and what justifies this choice of approach (despite its simplicity)?
Response: The above arguments are totally correct and the use of implicit solvent model is less reliable than the explicit solvent model, because of the use of real solvent molecules and considering real solvent-solute interactions in the explicit solvent model. However, as you mentioned (in the phrase “(despite its simplicity)”) the use of explicit solvent model is so time consuming and suffers much extra costs to the work. In fact, we think than the modelling of real solvated systems for these structures and complexes need a separate and independent work.
Additionally, several issues must be improved including thorough language editing, subscripts in SO2 and H2S, 'omega' instead of 'w', and capital letters in the name of omega-B97X-D functional (line 71 -- also the basis set symbol should be unified with line 108), E_ad in line 160-162 with the subscript. Also, the bibliography needs to be corrected (for instance ref. 24 and 25 involves Structural Chemistry rather than Journal of Structural Chemistry and in ref. 24 the year, volume, and pages should be corrected).
Response: Thanks. All of the above issues were corrected and modified carefully and completely.
Therefore, I would recommend reconsidering the manuscript for publication after major revision.
Reviewer 4 Report
The reviewed manuscript concerns the results obtained for doped carbon nanotubes with H2S, SO2 and thiophene, for which the theoretical calculations have been performed. I have the following critical remarks concerning this work:
- I suggest that in the title of publication, “DFT studies” instead of “DFT study” should be used.
- In Abstract should be “carbon nanotubes” instead of “CNTs”, an abbreviation with full name is given in Introduction, and next an abbreviation could be used.
- In the title of publication, Abstract and the whole manuscript, “2” in “H2S” and “SO2” should be given in subscript.
- On p. 3 (lines 104-107) is “Then, six doped structures containing one doped atom of aluminum (AN), boron (BN), sulfur (SN) , nitrogen (NN), phosphorus (PN), and silicon (SiN) were made by the replacement of one carbon atom with the heteroatom.”, but should be “Then, six doped structures containing one doped atom of aluminum (AN), boron (BN), nitrogen (NN), phosphorus (PN), sulfur (SN), and silicon (SiN) were made by the replacement of one carbon atom with the heteroatom.”, because this order is used in tables and figures.
- In the title of Table 1 should be given information about units, e. g. (in Å).
- On p. 4 (lines 133-135) is “It should be mentioned that in SO2-containing systems, the C-X values were maximum, while thiophene-containing systems owned the minimum C-X values.”, but I think that it’s no true (I checked it with values from Table 1).
- In the title of the first column in Tables 4 and 5 should be given information what is in this column, e. g. complex.
- Plots in Figure 3 should be increased.
- References should be unified according to Guide for Authors.
- Editorial mistakes should be corrected e.g.
- on p. 5 (line 141) is “2.996 Å respectively”, but should be “2.996 Å, respectively”,
- on p. 6 (lines 160, 161 and 162) and p. 7 (line 171) “ad” in “Ead” should be given in subscript.
- in Table 2 is “Thermal correction” and “Thermal Correction”. It should be unified,
- in Table 5, “g” in Eg” should be given in subscript.
- English should be carefully checked, e.g.
- in the Abstract (line 12) is “DFT calculations were used to calculate”, but should be “DFT method was used to calculate”,
- on p. 2 (line 73) is “wb97xd method is a DFT method”, but should be “wb97xd functional is a DFT method”,
- on p. 8 (line 215) is “Figures 3”, but should be “Figure 3”,
- on p. 10 (line 231) is “were this order:”, but should be “were in this order:”,
- on p. 10 (line 241) and Table 5 is “E(HOMO)”, but should be “E (HOMO)”,
- on p. 7 (line 176) and p. 10 (line 263) is “NBO calculations were used to calculate”, but should be “NBO program was used to calculate”.
According to mentioned above remarks, I suggest that in this paper, the minor revision is needed before publication in Quantum Reports.
Author Response
Dear referee
Thanks for the comments on our manuscript (Manuscript ID: quantumrep-1242622). Following, you may find our responses to the points requested more explanations/corrections. We hope the present format of the manuscript will be suitable for publishing.
Best Regards
Hossein Tavakol
Comments and Suggestions for Authors
The reviewed manuscript concerns the results obtained for doped carbon nanotubes with H2S, SO2 and thiophene, for which the theoretical calculations have been performed. I have the following critical remarks concerning this work:
- I suggest that in the title of publication, “DFT studies” instead of “DFT study” should be used.
Response: It was modified.
- In Abstract should be “carbon nanotubes” instead of “CNTs”, an abbreviation with full name is given in Introduction, and next an abbreviation could be used.
Response: It was corrected.
- In the title of publication, Abstract and the whole manuscript, “2” in “H2S” and “SO2” should be given in subscript.
Response: All of them were corrected.
- On p. 3 (lines 104-107) is “Then, six doped structures containing one doped atom of aluminum (AN), boron (BN), sulfur (SN) , nitrogen (NN), phosphorus (PN), and silicon (SiN) were made by the replacement of one carbon atom with the heteroatom.”, but should be “Then, six doped structures containing one doped atom of aluminum (AN), boron (BN), nitrogen (NN), phosphorus (PN), sulfur (SN), and silicon (SiN) were made by the replacement of one carbon atom with the heteroatom.”, because this order is used in tables and figures.
Response: This sentence was corrected as mentioned above.
- In the title of Table 1 should be given information about units, e. g. (in Å).
Response: They were added.
- On p. 4 (lines 133-135) is “It should be mentioned that in SO2-containing systems, the C-X values were maximum, while thiophene-containing systems owned the minimum C-X values.”, but I think that it’s no true (I checked it with values from Table 1).
Response: Sorry. That sentence was deleted.
- In the title of the first column in Tables 4 and 5 should be given information what is in this column, e. g. complex.
Response: The appropriate titles were added.
- Plots in Figure 3 should be increased.
Response: This figure was completely removed because of the request of the referee.
- References should be unified according to Guide for Authors.
Response: OK. They were carefully revised and unified according to Guide for Authors.
- Editorial mistakes should be corrected e.g.
- on p. 5 (line 141) is “2.996 Å respectively”, but should be “2.996 Å, respectively”,
- on p. 6 (lines 160, 161 and 162) and p. 7 (line 171) “ad” in “Ead” should be given in subscript.
- in Table 2 is “Thermal correction” and “Thermal Correction”. It should be unified,
- in Table 5, “g” in Eg” should be given in subscript.
Response: All of them were thoroughly corrected.
- English should be carefully checked, e.g.
- in the Abstract (line 12) is “DFT calculations were used to calculate”, but should be “DFT method was used to calculate”,
- on p. 2 (line 73) is “wb97xd method is a DFT method”, but should be “wb97xd functional is a DFT method”,
- on p. 8 (line 215) is “Figures 3”, but should be “Figure 3”,
- on p. 10 (line 231) is “were this order:”, but should be “were in this order:”,
- on p. 10 (line 241) and Table 5 is “E(HOMO)”, but should be “E (HOMO)”,
- on p. 7 (line 176) and p. 10 (line 263) is “NBO calculations were used to calculate”, but should be “NBO program was used to calculate”.
Response: Thanks for these corrections. All of the above mistakes were corrected. Moreover, the whole manuscript was carefully revised again to reduce language errors, as much as possible.
According to mentioned above remarks, I suggest that in this paper, the minor revision is needed before publication in Quantum Reports.
Round 2
Reviewer 1 Report
The revisions are fine. No additional comments.
Author Response
Comments and Suggestions for Authors
The revisions are fine. No additional comments.
Response: Thanks for your acceptance. The manuscript was revised again to reduce the language errors.
Reviewer 2 Report
Considering that this research's key flaw was just not explicitly stated part of the methods and all changes made to the manuscript, I think that the work can be accepted as is.
Author Response
Comments and Suggestions for Authors
Considering that this research's key flaw was just not explicitly stated part of the methods and all changes made to the manuscript, I think that the work can be accepted as is.
Response: Thanks for your acceptance.
Reviewer 3 Report
My only concern at this stage is that the title of subsection 3.4 Population analyses is misleading. Population analysis is provided in section 3.3 (entitled NBO calculations) with all the charges at tube and adsorbate. Section 3.4 is rather 'Reactivity parameters' since no charges are discussed here. This title must be modified not to confuse the reader.
Author Response
Comments and Suggestions for Authors
My only concern at this stage is that the title of subsection 3.4 Population analyses is misleading. Population analysis is provided in section 3.3 (entitled NBO calculations) with all the charges at tube and adsorbate. Section 3.4 is rather 'Reactivity parameters' since no charges are discussed here. This title must be modified not to confuse the reader.
Response: The title of this section was changed.